# Prevalence of Anemia in Chinese Children and Adolescents and Its Associated Factors

**DOI:** 10.3390/ijerph16081416

**Published:** 2019-04-19

**Authors:** Jinghuan Wu, Yichun Hu, Min Li, Jing Chen, Deqiao Mao, Weidong Li, Rui Wang, Yanhua Yang, Jianhua Piao, Lichen Yang, Xiaoguang Yang

**Affiliations:** Key Laboratory of Trace Element Nutrition of National Health Commission of the People’s Republic of China, Department of Trace Element Nutrition, National Institute for Nutrition and Health, Chinese Center for Disease Control and Prevention, No. 29 Nan Wei Road, Xicheng District, Beijing 100050, China; jhwu2012@163.com (J.W.); huyc@ninh.chinacdc.cn (Y.H.); limin@ninh.chinacdc.cn (M.L.); chenjing@ninh.chinacdc.cn (J.C.); maodq@ninh.chinacdc.cn (D.M.); liwd@ninh.chinacdc.cn (W.L.); wangrui@ninh.chinacdc.cn (R.W.); yangyh@ninh.chinacdc.cn (Y.Y.); piao_jianhua@sohu.com (J.P.)

**Keywords:** anemia, hemoglobin, Chinese children and adolescents, nutrition survey

## Abstract

In this study, we assessed the hemoglobin levels and anemia status of Chinese children and adolescents from the Chinese National Nutrition and Health Survey (CNNHS) in 2010–2012 and analyzed the factors associated with anemia. The hemoglobin concentration and prevalence of anemia for children and adolescents aged 6–17 years from both CNNHS 2010–2012 and CNNHS 2002 were analyzed. Multi-variable logistic regression analysis was used to assess the factors associated with anemia. The mean hemoglobin concentration increased among Chinese children and adolescents, from 135.2 ± 13.9g/L in CNNHS 2002 to 141.2 ± 15.8 g/L in CNNHS 2010–2012, with the prevalence of anemia decreasing from 12.6% to 6.6% (*p* < 0.0001). Anemia was specifically related to girls (*p* < 0.0001); children aged 6–8 years (*p* = 0.0175), 12–14 years (*p* = 0.0007), and 15–17 years (*p* < 0.0001); ordinary rural areas (*p* = 0.0009) and poor rural areas (*p* < 0.0001);spring (*p* < 0.0001), autumn (*p* < 0.0001), and winter (*p* < 0.0001);underweight individuals (*p* < 0.0001); and an annual average income per capita of less than 20,000 RMB (*p* < 0.0001).The prevalence of anemia in Chinese children and adolescents has improved significantly in comparison to 10 years prior; however, it remains a public health problem in this population. Further research is required to understand the determinants of iron status, which could then lead to strategies to alleviate iron deficiency for Chinese children and adolescents, especially for girls, those living in rural areas, underweight individuals, and those with a low family income.

## 1. Introduction

Anemia refers to a condition in which the number of red blood cells or their oxygen-carrying capacity is insufficient to meet physiological needs [1,2]. Anemia adversely impacts health and social economic development; children and women are particularly vulnerable [1,2]. In children, anemia may detrimentally affect cognitive development and physical growth from infancy to adolescence [3] and is associated with increased morbidity [4]. Anemia continues to be an important public health concern globally. The world wide anemia prevalence in 2010 was 32.9% in children and adults [5]; it is considerably more prevalent in developing than in developed countries [6].

Iron deficiency is estimated to be the most common cause of anemia worldwide [7]. In 2013, 1.2 billion people suffered from iron-deficiency anemia [8]. Other causes of anemia include parasitic infections, other nutritional deficiencies, such as folate, vitamin B_12_, and vitamin A deficiencies, chronic inflammation, and inherited disorders [7]. Hemoglobin levels can also vary greatly according to age, sex, altitude, smoking, and pregnancy status [9].

Much research has been reported on the prevalence of anemia worldwide; much of it focused on children under five years old (0–59 months) or pregnant women [6]; however, anemia data from children aged above five years and adolescents are limited. In China, although a few studies have investigated hemoglobin concentration and the prevalence of anemia in children and adolescents, they were based on relatively small sample sizes or regional compositions with specific populations, which made the studies unrepresentative [10,11,12].

The CNNHS has been administered every 10 years since 1982 to assess the nutritional status of Chinese citizens, and anemia is one of the important focuses of the survey. Hemoglobin concentration is used to evaluate anemia status in the CNNHS. According to the 2002 CNNHS, the prevalence of anemia was 20.1% for Chinese residents [13]. Since the late 2002s, the Chinese economy has undergone rapid development [14]; meanwhile, people’s dietary patterns have changed drastically [15]. In 2010–2012, China carried out its fourth National Nutrition and Health Survey. Using data from CNNHS 2002 and CNNHS 2010–2012, we aimed to (1) evaluate the hemoglobin concentration and the prevalence of anemia in children and adolescents aged 6–17 years, (2) compare the hemoglobin concentration and the prevalence of anemia among this population in 2002 and 2012, and (3) investigate the potential factors associated with anemia.

## 2. Materials and Methods

### 2.1. Study Design and Participants

Data of children and adolescents aged 6–17 years were obtained from CNNHS 2010–2012, which was the same survey as that reported in Li et al. [16] and Hu et al. [17]. This survey was a national, representative, and cross-sectional study administered to assess the health and nutrition status of Chinese citizens. It covered 31 provinces, autonomous regions, and municipalities directly under the central government throughout China (with the exceptions of Taiwan, Hong Kong, and Macao). A random stratified multistage cluster sampling design was used to recruit the participants of this survey. There were four strata in CNNHS 2010–2012: Large cities, small and medium-sized cities, ordinary rural areas, and poor rural areas, according to their economic characteristics and social development. Large cities included provincial capitals and municipalities with a population of more than one million. Small and medium-sized cities were defined as downtown areas that excluded the large cities. Poor rural areas were key poverty-stricken counties that were designated according to the 2001–2010 National Rural Poverty Alleviation and Development Program [18]. Ordinary rural areas were counties that were excluded from the key poverty counties. According to the population proportion, in the first stage of sampling, 34 large cities, 41 small and medium-sized cities, 45 rural areas, and 30 poor rural areas were selected as investigation sites (150 in total). The second stage involved six village committees or neighborhood committees from each investigation site. The last stage included 75 households from each selected village or neighborhood committee [19].

Data were also obtained from CNNHS 2002, which used the same design as CNNHS 2010–2012. In this survey, all counties (districts) and administrative units (including counties, county-level cities and districts) from 31 provinces in mainland China were divided into six categories: Two types of cities (large cities and small and medium-sized cities) and four types of rural areas (classes one, two, three, and four). This CNNHS included a total of 132 investigation sites in 2002. The second stage was the same as CNNHS 2010–2012. Finally, 90 households from each selected village or neighborhood committee were selected.

### 2.2. Ethical Approval

The survey was conducted according to the Declaration of Helsinki guidelines. All procedures for the survey involving human subjects were ethically approved by the Ethics Committee of the Institute for Nutrition and Health, Chinese Center for Disease Control and Prevention (China CDC) with the file number of 2013-018. All participants in this survey signed an informed consent form. As to the children and adolescents under 18 years old, informed written consent was obtained from the participants’ parents prior to the start of the study.

### 2.3. Blood Sample and Hemoglobin Measurement

The National Institute for Nutrition and Health, China CDC established a national project workgroup to develop a unified survey and questionnaire and conducted the survey using unified methods and materials. Hemoglobin levels were measured in the CDC laboratory in the district (county) of the survey area using the cyanmethemoglobin method, which is the most reliable laboratory method for the quantitative determination of hemoglobin and serves as a reference for the comparison and standardization of other methods [7]. This method is recommended by the International Committee for Standardization in Hematology [20]. Laboratory operations staff participated in the uniform national team training and examination. After becoming qualified, they could participate in the hemoglobin measurement.

After an overnight fast of at least 10 h, peripheral whole venous blood samples were collected in ethylene diamine tetra-acetic acid (EDTA) tubes; then, a 10 µL anticoagulant whole blood sample in a 10 µL quantitative capillary tube (Drummond Scientific Company, Broomall, PA, USA) was collected specifically for the hemoglobin test. Each blood sample had two parallel determinations. The intra- and inter-assay coefficients of variation (CVs) in the analytical methods were 1.03–2.91% and 2.04–3.70%, respectively. Blind samples (high- and low-value hemoglobin) were measured and qualified before proceeding with fieldwork. After the official commencement of fieldwork, one quality control analysis was tested in every 30 samples.

### 2.4. Criteria of Anemia

Normal hemoglobin distributions vary with sex, age, altitude, and, in women, pregnancy and menstrual status. Therefore, the correct interpretation of hemoglobin requires the consideration of modulating factors when selecting appropriate cut-off points. The age-specific cut-off points for hemoglobin at sea level are as follows: (1) For children aged under five years and pregnant women: ≥110 g/L normal, <110 g/L anemia; (2) for children aged 5–11 years: ≥115 g/L normal, <115 g/L anemia; (3) for children aged 12–14 years and non-pregnant women above 15 years old: ≥120 g/L normal, <120 g/L anemia; and (4) for men above 15 years of age: ≥130 g/L normal, <130 g/L anemia [7]. Hemoglobin levels at various altitudes were calculated according to the recommendations of the World Health Organization (WHO) [7].

### 2.5. Variables

All information concerning participants, including their age, sex, ethnicity, latitude, and family income was collected from each investigation site and recorded into the unified CNNHS System Platform. An anthropometric measurement for the participants was recorded by trained staff. Body mass index (BMI) was calculated as weight (kg) divided by height (m) squared, and its classification was based on the BMI z-score for individuals aged 6–17 years [21]. The season was recorded according to the month of blood sample collection. The latitude was defined according to the boundaries of China’s Qinling Mountains and Huaihe River, which are recognized as the boundaries that divide the north and the south of China.

### 2.6. Data Check

The hemoglobin level data of each monitoring site were recorded into the unified software. The unified standard of the data check principle was mainly achieved in the following aspects: (1) Connecting the hemoglobin database to the basic information base to check for duplicate information; (2) checking whether the difference in the hemoglobin values of parallel samples was less than 20%; and (3) according to the absorbance value of the spectrophotometer, calculating the hemoglobin level and then comparing it with reported values. Any problematic records were rejected and returned to the monitoring site for re-inspection.

### 2.7. Statistical Analysis

The population figures released by the National Bureau of Statistics of China in 2009 were used as a standard population [22]. Hemoglobin concentration and anemia rate analyses were adjusted for sample weights and the clustered survey design [19]. All participants in this study were divided into different sub-groups such as age group, region type, ethnicity, etc., according to different hypothesized predictors for anemia status. Hemoglobin levels were expressed as the mean ± standard deviation (SD) and were compared using the Kruskal–Wallis test. Frequencies were presented as percentages (%) and 95% confidence intervals (CIs). The rates of anemia were compared by the Rao–Scott test. The Bonferroni test was applied to conduct post-hoc analysis. Multivariable logistic regression analysis (proc surveylogistic) was used to analyze the relationship between anemia and possible predictors (e.g., age, sex, region type, BMI, season, family incomes). The odds ratio (OR) and 95% CIs were determined using multivariable logistic regression models. All the statistical analyses were conducted by Statistical Analysis Systems 9.4 software (SAS Institute Inc., Cary, NC, USA). A two-tailed *p*-value < 0.05 was considered statistically significant.

## 3. Results

### 3.1. Participant Characteristics

The distribution of children and adolescents from this survey according to age group, sex, region type, ethnicity, latitude, season, BMI, and income level is presented in Table 1. There were 33,015 subjects that had their hemoglobin measured, including 16,721 boys (53.2%) and 16,294 girls (46.8%). The distribution of participants was similar in terms of region type, ethnicity, latitude, season, and income level. There was a significant difference in the distribution of children and adolescents in terms of their age group and BMI classification (*p* < 0.05). Post-hoc analysis showed that sex distribution in children aged 6–8 years was significantly different from those aged 9–11 years (*p* = 0.006) or 12–14 years (*p* = 0.0024). The distribution of children with normal weight was significantly different from those in other groups (*p* < 0.0001).

### 3.2. Hemoglobin Concentration of Children and Adolescents

The hemoglobin levels for all participants in different sub-groups are shown in Table 2. The mean hemoglobin of all participants was 141.2 ± 15.8 g/L. Boys had a significantly higher concentration of hemoglobin (144.0 ± 16.3 g/L) than that of girls (138.0 ± 14.4 g/L, *p* < 0.0001). Significant differences were found in different age groups, region types, latitudes, seasons, BMI levels, and annual family income. Post-hoc analysis showed that participants aged 15–17 years had the highest level of hemoglobin (*p* < 0.0001). Children living in poor rural areas had the lowest levels of hemoglobin (*p* < 0.0001). The hemoglobin level in winter was significantly higher than that in autumn (*p* < 0.0001), spring (*p* < 0.0001), or summer (*p* < 0.0001). Obese children and adolescents had significantly higher hemoglobin levels than those with normal weight (*p* < 0.0001) or those underweight (*p* < 0.0001). Children with annual family incomes between 30,000 and 40,000 RMB had significantly higher hemoglobin levels than those with annual family incomes less than 10,000 RMB (*p* = 0.0003). However, there was no significant difference between the Han population and other populations (*p* = 0.3462). A similar tendency was found in terms of the age group, region type, ethnicity, latitude, season, BMI, and income level in boys and girls.

### 3.3. Anemia Prevalence for Children and Adolescents

As presented in Table 3, the prevalence of anemia in Chinese children and adolescents in 2010–2012 was 6.6%. Girls had a significantly higher prevalence of anemia (7.4%) than boys(6.0%, *p*<0.001). The prevalence of anemia varied among age groups, and the highest levels of anemia were found in adolescents aged 15–17 years (8.6%). Participants from large cities had the lowest prevalence of anemia (4.8%), and those from poor rural areas had the highest prevalence (9.5%). With respect to ethnicity, other populations (7.8%) had a significantly higher prevalence than the Han population (6.5%, *p* = 0.0096). Anemia was more prevalent among participants living in the north than those living in the south (*p* = 0.0106). In spring and summer, the prevalence of anemia was above 10.0%, which was significantly higher than that in autumn and winter (all *p*-values < 0.0001). Obese children and adolescents had the lowest prevalence of anemia (4.3%). The prevalence of anemia was the highest for participants with an annual income less than 10,000 RMB (7.5%), and the lowest for those with an annual income above 40,000 RMB. A similar tendency was found in both sexes in terms of their age group, region type, ethnicity, latitude, season, and income level.

### 3.4. Comparison of Anemia Status for Children and Adolescents between CNNHS 2010–2012 and CNNHS2002

As presented in Table 4, hemoglobin concentrations markedly increased over the decade, from 135.2 ± 13.9 g/L in 2002 to 141.2 ± 15.8 g/L in 2010–2012 (*p* < 0.001). This remarkable upward trend was similar in each sub-group, including different age groups, sexes, ethnicities, and region types. In addition, the prevalence of anemia significantly decreased over the same time span, from 12.6% in 2002 to 6.6% in 2010–2012 (*p* < 0.001). This clear downward trend was found in each sub-group, including different age groups, sexes, ethnicities, and region types. Among them, the prevalence of anemia in the four different age groups, minorities, and rural areas declined by more than half. 

### 3.5. Association between Anemia and Relative Variables

Factors including sex, age, region type, ethnicity, latitude, season, BMI level, and annual income were examined using a multi-variable logistic regression analysis for participants in CNNHS 2010–2012. The total number of participants was 32,714. As shown in Table 5, anemia was significantly associated with the female sex (OR = 1.275, *p* < 0.0001, relative to boys) and ages of 6–8 years and 12–17 years (ages 6–8 years, OR = 1.309, *p* = 0.0175; ages 12–14 years, OR = 1.676, *p* = 0.0007; ages 15–17 years, OR = 2.065, *p* < 0.0001; relative to those aged 9–11 years). Participants from ordinary rural areas or poor rural areas displayed an increased risk of anemia; the ORs were 1.205 (*p* = 0.0009) and 2.182 (*p* < 0.0001), respectively. Taking autumn as the reference standard, other seasons’ risks of anemia were increased, and the OR was 3.155 for spring (*p* < 0.0001), 2.819 for summer (*p* < 0.0001) and 1.091 for winter (*p* < 0.0001). An increased risk of anemia was observed in underweight children (OR = 1.741, *p* < 0.0001, relative to a BMI level above 28 kg/m^2^). Participants whose annual income was less than 20,000 RMB had an increased risk of anemia; with respect to an income level of more than 40,000 RMB, the OR was 1.704 for less than 10,000 RMB (*p* = 0.0001) and 1.608 for 10,000–19,999 RMB (*p* = 0.0099). However, no significant difference was observed in the risk of anemia according to ethnicity or latitude.

## 4. Discussion

In this study, we showed that the mean hemoglobin concentration of Chinese children and adolescents in CNNHS 2010–2012 was 141.2 g/L, which was a marked improvement compared with CNNHS 2002 findings. A significant decrease in the prevalence of anemia was observed from 12.6% in CNNHS 2002 to 6.6% in CNNHS 2010–2012. On the whole, the prevalence of anemia in Chinese children and adolescents represented only a mild public health problem during 2010–2012 according to the WHO criteria [23]. The improvement was in accordance with a recent study conducted by Song et al., who assessed the trend in sex disparity in hemoglobin concentration and the prevalence of anemia among Chinese school-aged children from 1995 to 2010 [24]. They found that the mean hemoglobin concentration increased among Chinese school-aged children between 1995 and 2010, from 132.7 g/L to 138.3 g/L in boys, and from 127.7g/L to 132.3 g/L in girls. The prevalence of anemia decreased from 18.8% in 1995 to 9.9% in 2010. However, such data were obtained from a different survey, the Chinese National Survey on Students Constitution and Health (CNSSCH), whose participants were all Han children aged 7, 9, 12, 14, and 17 years old, and samples of capillary blood were collected from the fingertip of each child. These differences might have produced the difference in the hemoglobin concentration values and the prevalence of anemia between CNNHS 2010–2012 and CNSSCH 2010.

We found that hemoglobin concentration and the prevalence of anemia varied with sex, age group, region type, season, BMI level, and annual income. The mean hemoglobin concentration of boys was significantly higher than that of girls, so the prevalence of anemia for boys was significantly lower than that for girls. Girls lose blood during menstruation when they enter puberty and thus are more vulnerable to anemia than boys. The difference between sexes in hemoglobin concentration and the prevalence of anemia was also observed in previous studies [11,24,25]. However, some studies reported no significant difference in the mean hemoglobin concentration between boys and girls, and the reason for this could be the different sample sizes and subjects [26,27].

Compared with those living in cities or ordinary rural areas in CNNHS 2010–2012, children and adolescents living in poor rural areas showed the highest prevalence and were at a greater risk of anemia. Although the Chinese economy has grown rapidly in both urban and rural areas [14], living conditions in urban areas remain better than those in rural areas [28,29]. Thus, the nutritional status of people in cities is often better than those in rural areas. Compared with the Han population, ethnic minorities had a higher prevalence of anemia. In China, ethnic minorities often live in remote regions, and their living standards are not as good as those of the Han population. Song et al. found that the prevalence of anemia was different among students of different ethnic minorities [12]. The prevalence of anemia for Hui ethnic students was a moderate public health problem according to the WHO criteria [12]. Although the effects of seasonal changes on anemia have rarely been reported, they may be an important factor. The results of the current study indicate that the highest prevalence of anemia occurs in spring, followed by summer. Ronnenberg et al. reported that distinct seasonal trends were observed in the prevalence of moderate anemia for Chinese women of a childbearing age [29,30]. The hemoglobin concentrations of these women were significantly lower in summer than those in other seasons [30]. Chinese people often have a strong sense of strengthening their resistance to a cold winter by consuming more red meat in autumn, whereas in spring and summer, people often have less of an appetite because of the hot weather.

Overweight and obese children had a significantly lower prevalence of anemia than those with normal or low body weight. This is in accordance with reports on Chinese people by He et al. and Qin et al. [31,32]. Bentley et al. also found that overweight women were at a lower risk of anemia than those with a normal or low BMI in India, which could be due to a better dietary nutritional intake in overweight or obese people [33]. However, some studies found that obesity is significantly associated with iron deficiency, and could be caused by obesity-related inflammation and/or related comorbidities [34,35]. The disparity among those findings might be due to the different dietary patterns in different regions. The relationship between BMI and anemia needs to be further researched. Family income also had an impact on anemia. The highest prevalence of and vulnerability to anemia was observed in children whose family income was less than 10,000 RMB per year. This could be explained by the fact that their parents may have less ability and awareness to improve their children’s nutrition due to their poor socioeconomic and household conditions. Previous studies demonstrated associations between anemia and indicators of poor socioeconomic conditions in China [36] and in other developing countries, such as parts of Africa [37,38]. Li et al. found a negative correlation between household wealth and the prevalence of anemia in Central and Eastern China [11]. Ngnie-Teta et al. [37] and Custodio et al. [38] found that children who lived in households with lower living standards or lower socio-educational levels were at greater risk of anemia.

However, considerable improvement has occurred in terms of hemoglobin concentration and the prevalence of anemia in different age groups, sexes, ethnicities, and regions in Chinese children and adolescents since 2002. Efforts have been made to decrease the prevalence of anemia in China. Firstly, some specific nutritional policies implemented by the government have played an important role in alleviating anemia, such as The Standard Amount of Nutritional Supply for Student Lunch issued by the Ministry of Health in 1998 [39], and The Nutrition Improvement Program for Rural Compulsory Education Students issued by the State Council in October 2011 [40]. As the population at the highest risk of anemia, rural school-aged children, especially rural girls, may have benefitted from these policies. Secondly, public health strategies contributed to preventing and improving resistance to anemia, such as food fortification, micronutrient supplementation, nutritional and public health awareness enhancement, and the improvement in dietary diversity [41]. Thirdly, with the growing economy, the living standards of Chinese citizens have been improved, providing more opportunity for people to consume more iron-rich foods.

There were some limitations to this study. Firstly, the iron status of children was not examined; thus, it is not possible to speculate about the causal relationship between iron and anemia in children and adolescents. The survey did not include some causes of improvement in anemia rates, such as iron supplements or other food fortifications. Lastly, this survey did not consider that shifting the nutrient status for a whole population may lead to some people being exposed to excess levels of a nutrient. Further surveys need to pay more attention to these aspects to avoid these limitations.

## 5. Conclusions

In summary, hemoglobin levels and prevalence of anemia have improved among Chinese children and adolescents between 2002 and 2012. However, the prevalence of anemia remained relatively high for girls, 15–17-year-old youths, rural areas, and in households with a low income. Children and adolescents aged 6–17 years are at an important stage of physical and mental development. Further research is required to understand the determinants of iron status, which could then lead to strategies to alleviate iron deficiency for Chinese children and adolescents, especially for girls, those living in rural areas, underweight individuals, and those with a low family income.

## Figures and Tables

**Table 1 ijerph-16-01416-t001:** Demographic characteristics of children and adolescents from the CNNHS 2010–2012.

Variable	Total (N)	Boys (N)	Girls (N)	*p*-Value ^a^
Age group (years)				0.0150
6–8	7903	4011	3892	
9–11	8689	4411	4278	
12–14	8794	4456	4338	
15–17	7629	3843	3786	
Region type				0.2036
Large cities	6859	3455	3404	
Small and medium-sized cities	9729	4912	4817	
Ordinary rural areas	10,501	5352	5149	
Poor rural areas	5926	3002	2924	
Ethnicity				0.0666
Han population	29,699	15,069	14,630	
Other population	3054	1498	1556	
Latitude				0.7182
North	14,943	7544	7399	
South	18,072	9177	8895	
Season				0.8778
Spring	2227	1135	1092	
Summer	1786	899	887	
Autumn	17,801	9044	8757	
Winter	10,938	5488	5450	
BMI classification				<0.0001
Underweight	2884	1720	1164	
Normal	24,602	11,635	12,967	
Overweight	3307	1959	1348	
Obesity	2222	1407	815	
Income (RMB) ^b^				0.2139
<10,000	12,607	6361	6246	
10,000–19,999	8066	4151	3915	
20,000–29,999	2794	1453	1341	
30,000–39,999	1206	609	597	
≥40,000	1055	531	524	
No response	6987	3440	3547	

**^(a)^***p*-value for Rao–Scott test; **^(b)^** annual average income per capita; BMI: Body mass index.

**Table 2 ijerph-16-01416-t002:** Hemoglobin concentration of children and adolescents from CNNHS 2010–2012.

Variable	Total (g/L)	*p*-Value ^a^	Boys(g/L)	*p*-Value ^b^	Girls(g/L)	*p*-Value ^c^
Total	141.2 ± 15.8		144.0 ± 16.3		138.0 ± 14.4	
Age group (years)		<0.0001		<0.0001		<0.0001
6–8	135.5 ± 13.4		135.6 ± 12.9		135.4 ± 13.9	
9–11	138.7 ± 13.5		138.8 ± 13.4		138.7 ± 13.6	
12–14	142.3 ± 15.8		145.2 ± 15.9		138.9 ± 14.9	
15–17	146.7 ± 17.1		153.5 ± 16.0		138.9 ± 14.8	
Region type		<0.0001		<0.0001		<0.0001
Large cities	142.4 ± 15.2		146.0 ± 15.6		138.6 ± 13.8	
Small and medium-sized cities	141.0 ± 16.1		144.0 ± 16.6		137.6± 14.8	
Ordinary rural areas	142.0 ± 15.5		144.8 ± 16.2		138.8 ± 14.0	
Poor rural areas	139.8 ± 15.5		141.8 ± 16.1		137.4 ± 14.3	
Ethnicity		0.3462		0.1543		0.6570
Han population	141.2 ± 15.7		144.0 ± 16.4		138.0 ± 14.3	
Other population	141.4 ± 16.0		144.4 ± 16.4		138.2 ± 14.9	
Latitude		<0.0001		<0.0001		0.3373
North	141.6 ±16.0		144.5 ± 16.4		138.2 ± 14.9	
South	140.9± 15.5		143.6 ± 16.3		137.9 ± 14.0	
Season		<0.0001		<0.0001		<0.0001
Spring	138.8 ± 20.6		141.7 ± 21.2		135.5 ± 19.4	
Summer	138.3 ± 17.7		140.7 ± 17.8		135.4 ± 17.3	
Autumn	140.9 ± 14.8		143.5 ± 15.6		137.8 ± 13.1	
Winter	142.9 ± 15.8		146.0 ± 16.1		139.3 ± 14.6	
BMI classification		<0.0001		<0.0001		0.0055
Underweight	140.2 ± 16.4		142.0 ± 16.7		137.1 ± 15.5	
Normal	141.2 ± 15.6		144.2 ± 16.3		138.1 ± 14.2	
Overweight	141.8 ± 16.3		144.3 ± 16.8		138.0 ± 14.7	
Obesity	142.4 ± 15.5		144.5 ± 15.2		138.5 ± 15.2	
Income (RMB) ^d^		<0.0001		<0.0001		0.1980
<10,000	140.8 ± 15.9		143.2 ± 16.7		138.1 ± 14.5	
10,000–20,000	141.3 ± 15.6		144.2 ± 16.0		138.0 ± 14.6	
20,000–30,000	142.3 ± 15.9		145.4 ± 16.7		138.6 ± 13.9	
30,000–40,000	142.9 ± 13.9		145.4 ± 14.3		140.0 ± 12.7	
≥40,000	141.9 ± 14.6		145.8 ± 14.6		137.7 ± 13.4	
No response	141.1 ± 16.0		144.6 ± 16.5		137.4 ± 14.6	

Values are means ± SD; **^(a–c)^**
*p*-values for Kruskal–Wallis test; **^(d)^** annual average income per capita; BMI: Body mass index.

**Table 3 ijerph-16-01416-t003:** Anemia prevalence for children and adolescents from CNNHS 2010–2012.

Variable	Total (%(95%CI))	*p*-Value ^a^	Boys (%(95%CI))	*p*-Value ^b^	Girls (%(95%CI))	*p*-Value ^c^
Total	6.6 (6.3–6.9)		6.0 (5.6–6.3)		7.4 (6.9–7.8)	
Age group (years)		<0.0001		<0.0001		<0.0001
6–8	5.7 (5.2–6.3)		5.2 (4.5–6.0)		6.3 (5.5–7.1)	
9–11	4.3 (3.9–4.7)		4.2 (3.5–4.8)		4.5 (3.8–5.1)	
12–14	7.2 (6.7–7.8)		5.9 (5.1–6.6)		8.9 (8.0–9.8)	
15–17	8.6 (7.9–9.2)		8.0 (7.1–8.9)		9.2 (8.2–10.2)	
Region type		<0.0001		<0.0001		0.0003
Large cities	4.8 (4.3–5.3)		3.4 (2.8–4.0)		6.4 (5.6–7.2)	
Small and medium–sized cities	6.5 (6.0–7.0)		5.7 (5.1–6.3)		7.5 (6.7–8.2)	
Ordinary rural areas	5.8 (5.3–6.2)		5.0 (4.4–5.6)		6.6 (5.9–7.3)	
Poor rural areas	9.5 (8.7–10.2)		9.8 (8.7–10.9)		9.1 (8.0–10.1)	
Ethnicity		0.0096		0.5310		0.0033
Han population	6.5 (6.2–6.8)		5.9 (5.5–6.3)		7.2 (6.7–7.6)	
Other population	7.8 (6.8–8.8)		6.4 (5.0–7.7)		9.4 (7.8–10.9)	
Latitude		0.0106		0.0965		0.0523
North	7.0 (6.6–7.5)		6.3 (5.7–6.9)		7.8 (7.2–8.5)	
South	6.2 (5.9–6.6)		5.6 (5.1–6.2)		6.9 (6.3–7.5)	
Season		<0.0001		<0.0001		<0.0001
Spring	15.1 (13.3–16.8)		13.7 (11.4–16.1)		16.6 (14.1–19.2)	
Summer	13.2 (11.4–14.9)		11.2 (8.9–13.5)		15.5 (12.8–18.1)	
Autumn	5.4 (5.1–5.8)		5.1 (4.6–5.6)		5.8 (5.3–6.4)	
Winter	5.8 (5.3–6.3)		5.0 (4.3–5.7)		6.8 (6.0–7.5)	
BMI classification		<0.0001		<0.0001		0.0793
Underweight	8.9 (7.8–10.0)		8.6 (7.2–10.0)		9.4 (7.6–11.2)	
Normal	6.6 (6.2–6.9)		5.9 (5.5–6.4)		7.2 (6.7–7.7)	
Overweight	6.3 (5.4–7.3)		5.6 (4.4–6.7)		7.5 (5.9–9.1)	
Obesity	4.3 (3.4–5.3)		3.2 (2.2–4.2)		6.4 (4.5–8.3)	
Income (RMB) ^d^		<0.0001		<0.0001		0.0415
<10,000	7.5 (7.0–7.9)		7.4 (6.7–8.0)		7.6 (6.9–8.3)	
10,000–20,000	6.6 (6.0–7.2)		5.4 (4.6–6.1)		8.0 (7.0–8.9)	
20,000–30,000	6.0 (5.1–7.0)		5.7 (4.3–7.0)		6.5 (5.0–8.0)	
30,000–40,000	4.0 (2.7–5.2)		3.5 (1.8–5.2)		4.5 (2.7–6.3)	
≥40,000	3.6 (2.3–4.9)		2.1 (0.6–3.5)		5.3 (3.2–7.5)	
No response	6.2 (5.6–6.8)		5.1 (4.3–6.0)		7.4 (6.4–8.3)	

Values are presented as % with 95% CI in parentheses; **^(a–c)^**
*p*-value for the Rao–Scott test; **^(d)^** annual average income per capita; BMI: Body mass index.

**Table 4 ijerph-16-01416-t004:** Comparison of hemoglobin concentration and anemia prevalence for children and adolescents between CNNHS 2010–2012 and CNNHS 2002.

Variable	N			Hb(g/L) ^a^			Prevalence (% (95% CI)) ^b^		
	2002	2010–2012	*p*-Value ^c^	2002	2010–2012	*p*-Value ^d^	2002	2010–2012	*p*-Value ^e^
Total	48,610	33,015		135.2 ± 13.9	141.2 ± 15.8	<0.0001	12.6 (12.3–12.9)	6.6 (6.3–6.9)	<0.0001
Normal	42,472	30,937	<0.0001	138.4 ± 11.3	143.4 ± 13.4	<0.0001	–	–	
Anemia	6138	2078		113.2 ± 9.7	110.1 ± 13.1	<0.0001	12.6 (12.3–12.9)	6.6 (6.3–6.9)	<0.0001
Age group (years)									
6–8	14,678	7903	<0.0001	132.2 ± 12.8	135.5 ± 13.4	<0.0001	11.8 (11.3–12.3)	5.7 (5.2–6.3)	<0.0001
9–11	16,020	8689		134.5 ± 12.8	138.7 ± 13.5	<0.0001	9.6 (9.1–10.1)	4.3 (3.9–4.7)	<0.0001
12–14	12,174	8794		137.0 ± 14.2	142.3 ± 15.8	<0.0001	14.8 (14.2–15.4)	7.2 (6.7–7.8)	<0.0001
15–17	5738	7629		140.9 ± 16.3	146.7 ± 17.1	<0.0001	18.5 (17.5–19.5)	8.6 (7.9–9.2)	<0.0001
Sex									
Boys	25,133	16,721	0.003	136.3 ± 14.5	144.0 ± 16.3	<0.0001	12.1 (11.7–12.5)	6.0 (5.6–6.3)	<0.0001
Girls	23,476	16,294		134.0±13.3	138.0 ± 14.4	<0.0001	13.1 (12.7–13.6)	7.4 (6.9–7.8)	<0.0001
Ethnicity									
Han	42,356	29,699	<0.0001	135.0 ± 13.7	141.2 ± 15.7	<0.0001	12.1 (11.7–12.4)	6.5 (6.2–6.8)	<0.0001
Other population	6254	3054		136.7 ± 15.2	141.4 ± 16.0	<0.0001	16.5 (15.5–17.4)	7.8 (6.8–8.8)	0.0001
Region type									
Urban	21,468	16,588	<0.0001	135.7 ± 13.5	141.1 ± 16.0	<0.0001	9.5 (9.1–9.8)	6.3 (5.9–6.7)	<0.0001
Rural	27,142	16,427		134.8 ± 14.3	141.3 ± 15.5	<0.0001	15.1 (14.7–15.6)	6.9 (6.5–7.3)	<0.0001

**^(a)^** Values are means ± SD; **^(b)^** values are presented as % (95%CI); **^(c)^**
*p*-value for the Rao-Scott; **^(d)^**
*p*-value for Kruskal–Wallis test; **^(e)^**
*p*-value for the Rao–Scott test.

**Table 5 ijerph-16-01416-t005:** Determinants of anemia among Chinese children and adolescents from CNNHS 2010–2012.

Variable	OR (95% CI) ^a^	*p*-Value ^b^
Sex		
Boys	ref	
Girls	1.275 (1.155,1.407)	<0.0001
Age group (years)		
9–11	ref	
6–8	1.309 (1.124,1.525)	0.0175
12–14	1.676 (1.453,1.932)	0.0007
15–17	2.065 (1.794,2.376)	<0.0001
Region type		
Large cities	ref	
Small and medium-sized cities	1.388 (1.209,1.593)	0.9167
Ordinary rural areas	1.205 (1.040,1.396)	0.0009
Poor rural areas	2.182 (1.869,2.548)	<0.0001
Ethnicity		
Han	ref	
Other population	0.894 (0.762,1.048)	0.1667
Latitude		
South	ref	
North	1.011 (0.918,1.114)	0.8164
Season		
Autumn	ref	
Spring	3.155 (2.711,3.671)	<0.0001
Summer	2.819 (2.387,3.33)	<0.0001
Winter	1.091 (0.966,1.231)	<0.0001
BMI classification		
Obesity	ref	
Underweight	1.741 (1.324,2.289)	<0.0001
Normal	1.302 (1.026,1.652)	0.6464
Overweight	1.380 (1.043,1.824)	0.6019
Income (RMB) ^c^		
≥40,000	ref	
<10,000	1.704 (1.168,2.485)	0.0001
10,000–19,999	1.608 (1.098,2.354)	0.0099
20,000–29,999	1.589 (1.056,2.391)	0.0875
30,000–39,999	1.058 (0.647,1.731)	0.0737
No response	1.425 (0.970,2.093)	0.5374

**^ (a)^** Values are presented as% (95% CI); **^(b)^**
*p*-value for multi-variable logistic regression analysis; **^(c)^** annual average income per capita; BMI: Body mass index.

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
