# Peer review of "Prevalence of Anemia in Chinese Children and Adolescents and Its Associated Factors"

_ijerph, 2019, doi:10.3390/ijerph16081416_

Round 1
Reviewer 1 Report
The manuscript is still confusing.
Lines 56-60 - the authors state the anemia prevalence for chinese resident except children under 6 in both the 2002 study and the 2010 study. If this data has already been published - why are you publishing the data here for 6-17 year old children.
Line 218 - the phrase "in contrast" is incorrect. If Hb increases we expect anaemia to decrease. Change"In contrast" "Also"
Line 330-332 - This is incorrect also. Intervention studies would be carried out if we had a hypothesis about how to improve iron status. It would be better to say further research is required to understand the determinants of iron status, which could then lead to strategies to alleviate iron deficiency.
Author Response
Response to Reviewer 1 Comments
Dear Reviewer:
Thank you very much for your comments on our paper submitted to IJERPH (Manuscript ID ijerph-482793).
We asked MDPI to help us edit English and they made revision carefully to our manuscript. We revised the manuscript in accordance with the reviewers’ comments, and carefully proof-read the manuscript to minimize typographical, grammatical, and other errors. Here below is our response to Reviewer 1 comments:
Point 1: The manuscript is still confusing.
Lines 56-60 - the authors state the anemia prevalence for chinese resident except children under 6 in both the 2002 study and the 2010 study. If this data has already been published - why are you publishing the data here for 6-17 year old children.
Response 1: Thanks reviewer for mentioning this important problem. We didn’t express clearly, and made the paper confusing. In both the CNNHS 2002 and CNNHS 2010–2012, the prevalence of anemia for Chinese residents except children under six years was 20.1% and 9.7% respectively, which were just for whole participants. The prevalence for children and adolescents aged 6-17 years old was not analyzed. Therefore, these data was not published. In avoid of misunderstanding, we deleted the confusing part and rewrote them, please see the line 58–59.
Point 2: Line 218 - the phrase "in contrast" is incorrect. If Hb increases we expect anaemia to decrease. Change"In contrast" "Also"
Response 2: The reviewer was very careful and found this significant problem. "In contrast" is incorrect and we changed it to "Also" according to reviewer’s suggestion.
Point 3: Line 330-332 - This is incorrect also. Intervention studies would be carried out if we had a hypothesis about how to improve iron status. It would be better to say further research is required to understand the determinants of iron status, which could then lead to strategies to alleviate iron deficiency.
Response 3: Thanks for the recommendation from the reviewer. We revised this part as” Further research is required to understand the determinants of iron status, which could then lead to strategies to alleviate iron deficiency for Chinese children and adolescents, especially for girls, those living in rural areas, underweight individuals, and those with a low family income. In addition, we revised the conclusion of the abstract. Please see the line 26–29 and the line 328–331.

Reviewer 2 Report
The manuscript has been markedly improved in every aspect. The language is clear and well written, statistical analysis robust and well explained, and findings interesting and supportive of the discussion.
A few minor points
line 52 - I'm not sure I understand what the authors mean by "incomprehensive"? not understandable? not covering all groups of interest?
line 62- if anemia status was "not assessed" how did the authors get the data to which they refer in line 68?
line 120 - pregnancy and menstrual status
line179 - what does the phrase "total hemoglobin" mean? total of what - every person in the study - ie the mean hgb of all participants?
lines 184-186 - where the seasonal differences found in every region and amongst all different groups/genders etc?
line 203 - how much "higher prevalence" specifically?
table 5 - how were p's calculated?
line 246 - this may simply be a matter of style preference - that is to say my preference - but perhaps "In this study we showed"? or We have demonstrated? The phrase is not exactly incorrect but slightly awkward
line 324 - suggest changing "on" to "to"
line 330 - intervention studies to evaluate - rather than prove - prove what?
Author Response
Response to Reviewer 2 Comments
Dear Reviewer:
Thank you very much for your comments on our paper submitted to IJERPH (Manuscript ID ijerph-482793).
We asked MDPI to help us edit English and they made revision carefully to our manuscript. We revised the manuscript in accordance with the reviewers’ comments, and carefully proof-read the manuscript to minimize typographical, grammatical, and other errors. Here below is our response to Reviewer 2 comments:
Point 1: line 52 - I'm not sure I understand what the authors mean by "incomprehensive"? not understandable? not covering all groups of interest?
Response 1: Thanks reviewer for mentioning this important problem. "incomprehensive" was confusing, we deleted it according to reviewer’s suggestion.
Point 2: line 62- if anemia status was "not assessed" how did the authors get the data to which they refer in line 68?
Response 2: The reviewer was very careful and found this significant problem. We didn’t express clearly, and made the paper confusing. In both the CNNHS 2002 and CNNHS 2010–2012, the prevalence of anemia for Chinese residents except children under six years was 20.1% and 9.7% respectively, which were just for whole participants. The prevalence for children and adolescents aged 6-17 years old was not analyzed. Therefore, these data was not published. The references in line 68 were used to state that the data of these studies were from the same survey, the CNNHS2010–2012. One of references assessed the prevalence of anemia for Chinese rural residents, and the other one assessed the Vitamin D nutritional status. So far, this paper was the first research to assess the prevalence of anemia for children and adolescents aged 6–17 years in CNNHS 2010–2012. In avoid of misunderstanding, we deleted the confusing part and rewrote them, please see the line 57–58 and the line 66.
Point 3: line 120 - pregnancy and menstrual status
Response 3: Thanks for the recommendation from the reviewer. The sentence has been revised according to reviewer’s suggestion.
Point 4: line179 - what does the phrase "total hemoglobin" mean? total of what - every person in the study - ie the mean hgb of all participants?
Response 4: The reviewer was very careful and found this important problem. "total hemoglobin" means “the mean hgb of all participants”. In avoid of misunderstanding, the sentence was changed to “The mean hemoglobin of all participants was 141.2±15.8 g/L.” according to reviewer’s suggestion.
Point 5: lines 184-186 - where the seasonal differences found in every region and amongst all different groups/genders etc?
Response 5: Thanks for the recommendation from the reviewer. Seasonal differences were just analyzed among the four seasons, and were not analyzed with other factors. Therefore, seasonal differences were not found in any region and amongst all different groups/genders etc.
Point 6: line 203 - how much "higher prevalence" specifically?
Response 6: The reviewer was very careful and found this important problem. The prevalence of anemia for other populations and the Han population were supplemented, and the sentence was revised to “For ethnicity, other populations (7.8%) had significantly higher prevalence than the Han population (6.5%, p=0.0096).”
Point 7: table 5 - how were p's calculated?
Response 7: p's in table 5 were calculated by multi-variable logistic regression analysis. As for each factor, the sub-group which had a lower prevalence of anemia was set as a reference.
Point 8: line 246 - this may simply be a matter of style preference - that is to say my preference - but perhaps "In this study we showed"? or We have demonstrated? The phrase is not exactly incorrect but slightly awkward
Response 8: Thanks for the recommendation from the reviewer. "In this study we showed" is much better than “We showed”, and we revised it according to reviewer’s suggestion.
Point 9: line 324 - suggest changing "on" to "to"
Response 9: Thanks for the recommendation from the reviewer. “on” was changed to “to” according to reviewer’s suggestion.
Point 10: line 330 - intervention studies to evaluate - rather than prove - prove what?
Response 10: Thanks for the recommendation from the reviewer. Reviewer 1 suggested also made some revision to this part. According to viewer’s suggestion, we changed them to “Further research is required to understand the determinants of iron status, which could then lead to strategies to alleviate iron deficiency for Chinese children and adolescents, especially for girls, those living in rural areas, underweight individuals, and those with a low family income.” In addition, we revised the conclusion of the abstract. Please see the line 26–29 and the line 328–331.

This manuscript is a resubmission of an earlier submission. The following is a list of the peer review reports and author responses from that submission.
Round 1
Reviewer 1 Report
This paper describes a study investigating haemoglobin concentrations in Chinese children and adolescents between 2010 and 2012 and shows how it has increased since 2002. This paper would be of interest to those working in this field.
It is not clear if the authors are reporting solely the data from children in the 2010-2012 survey or also from the children in 2002 survey. If the former, could the authors add a brief summary to the introduction. If the former, they will need to include the methods for the first survey to the methods section.
Introduction
Line 40 – change to iron deficiency anaemia.
Line 44 - change to ‘there is much research on…’
Line 53 - is the prevalence of 20.1% of anaemia in the adults population?? You need to make this clear.
Line 55 – again make it clear if this 9.7% is for adults.
Line 56-57. This is unclear, is the anemia status of school-aged children unclear for the 2002 and 2010-12 survey or just the 2010-12 survey. If the data has already been published for the 2002 survey could you put the level of anemia here with the reference.
Methods
How were participants contacted and actually recruited, this needs to be included in the methods. Was data collected on those who refused to take part, how might these differ to participants.
How many of your participants had haemoglobin levels above recommendations? If you have this data you need to report this.
Results
Line 143 – remove “around” give the exact figure.
Line 152-153 –This sentence is unclear, do not start a sentence with “except”. First state which had significant differences, then which did not.
Table 2 – why does the first row of the table - total have a p value, what does this refer to?
Line 183-186 – this sentence is tricky to follow – please simplify.
Discussion
In the limitations section, you also need to comment on the % with haemoglobin above recommendations. Shifting nutrient status up for a whole population can lead to some people being exposed to excess levels, you need to consider this.
Conclusion
Before an intervention can be carried out you first need to understand why iron status is still low in girls, so you can then target the intervention.
Author Response
Response to Reviewer 1 Comments
Dear Reviewer:
Thank you very much for your comments on our paper submitted to IJERPH (Manuscript ID ijerph-436940).
We revised the manuscript and carefully proof-read the manuscript to minimize typographical, grammatical, and bibliographical errors. Here below is our description on revision according to your comments:
Point 1: This paper describes a study investigating haemoglobin concentrations in Chinese children and adolescents between 2010 and 2012 and shows how it has increased since 2002. This paper would be of interest to those working in this field.
It is not clear if the authors are reporting solely the data from children in the 2010-2012 survey or also from the children in 2002 survey. If the former, could the authors add a brief summary to the introduction. If the former, they will need to include the methods for the first survey to the methods section.
Response 1: We mainly reported the data from children in the 2010-2012 survey, and then compared it with the 2002 survey. There was a brief summary on the survey in the introduction. The methods for the first survey were supplemented to the methods section according to the reviewer’s suggestion.
Point 2: Introduction: Line 40 – change to iron deficiency anaemia.
Response 2: It has been changed to” iron-deficiency anemia”
Point 3: Line 44 - change to ‘there is much research on…’
Response 3: It has been changed to ‘there is much research on…’
Point 4: Line 53 - is the prevalence of 20.1% of anaemia in the adults population?? You need to make this clear.
Response 4: We have made this clear that the prevalence of anemia was 20.1% in the 2002 CNNHS for the Chinese residents except children under 6 years old.
Point 5: Line 55 – again make it clear if this 9.7% is for adults.
Response 5: We have made this clear that the prevalence of anaemia was 9.7% in the 2010-2012 CNNHS for the Chinese residents except children under 6 years old.
Point 6: Line 56-57. This is unclear, is the anemia status of school-aged children unclear for the 2002 and 2010-12 survey or just the 2010-12 survey. If the data has already been published for the 2002 survey could you put the level of anemia here with the reference?
Response 6: The anemia status of school-aged children was unclear for the 2010-12 CNNHS survey and the 2002 CNNHS.
Point 7: Methods: How were participants contacted and actually recruited, this needs to be included in the methods. Was data collected on those who refused to take part, how might these differ to participants.
Response 7: Participants were contacted by the staff of Center for Disease Control and Prevention China (CDC) in each investigation site. They were actually recruited by a random stratified multistage cluster sampling design. The data used in this study did not collect on those who refused to take part.
Point 8: How many of your participants had haemoglobin levels above recommendations? If you have this data you need to report this.
Response 8: There were 30836 (33015×0.936) participants had haemoglobin levels above recommendations. We have reported this number in the result section according to reviewer’s suggestion.
Point 9: Results: Line 143 – remove “around” give the exact figure.
Response 9: “Around” was removed and the exact figure was given.
Point 10: The reviewer’s comment: Line 152-153 –This sentence is unclear, do not start a sentence with “except”. First state which had significant differences, then which did not.
Response 10: The sentence was revised. First stated which had significant differences, and then which did not.
Point 11: Table 2 – why does the first row of the table - total have a p value, what does this refer to?
Response 11: Different p values were explained in a legend at the bottom of the table. The first row of the table - total have a p value for significant tests among different subgroups.
Point 12: Line 183-186 – this sentence is tricky to follow – please simplify.
Response 12: The sentence was revised.
Point 13: Discussion: In the limitations section, you also need to comment on the % with haemoglobin above recommendations. Shifting nutrient status up for a whole population can lead to some people being exposed to excess levels, you need to consider this.
Response 13: The reviewer was very thoughtful and found this significant limitation. We considered this and added it to the limitations section.
Point 14: Conclusion: Before an intervention can be carried out you first need to understand why iron status is still low in girls, so you can then target the intervention.
Response 14: We revised the conclusion according to the suggestion.

Reviewer 2 Report
This is an interesting study on the prevalence of anaemia and measurement of haemoglobin levels in two cohorts of Chinese children, and it is encouraging to see the improvement in haemoglobin levels and the fall in anaemia prevalence found in the study. The authors have done a good job analysing factors that are associated with anaemia and thus possible implications for future public health policies. Following are my general comments and then specific points line by line after that.
There are places where the English language used is somewhat confusing and I have highlighted some of those areas in the comments below with possible corrections for the authors to consider, but I would suggest that a review from someone with an excellent command of the English language might be helpful.
The abstract needs work - some language issues but also I find there are more results presented than the significance of the findings (ie discussion).
I find the data tables difficult to interpret especially the interpretation of the p's -sometimes it is difficult to know what the comparison being made is, and also as different statistical tests were used it is not clear from where the p is derived - ie what statistical test yielded that p?
lastly, this is an observational study - thus although haemoglobin levels have improved and anaemia levels decreased causation has not been directly demonstrated - an intervention study is needed to prove that the public health policies rather than some other socio/cultural/economic factors had these effects - and furthermore which policies were most effective - this comment applies in particular to the authors' conclusion
Specific points:
line 15 - may I suggest "factors associated with anemia" - the statement as is sounds awkward
line 15- the authors use American English throughout the manuscript (for example, anemia) other than "haemoglobin" - for consistency they should use "hemoglobin" - this error occurs through-out the manuscript
line 16 - "between THE 2010, and after 48,610 - children and adolescents
line 19- 135.2 is missing units of measurement, and "with THE prevalence"
line 20 - is there a "p" for this difference?
line 32 - awkard language
line 40 - "iron-deficiency"
line 41 "OTHER nutritional deficiencies"
line 44 - "there are lots" sounds like slang
line 45 - no need for whilst at the start of the sentence
line 46 - please provide references to support these statements
line 49 - please provide references to support these statements
line 54 - please provide references to support these statements
line 55- please provide references to support these statements
line 57 - why did the anemia status remain unclear?
line 66 - please check grammar
line 73- what standards to define socioeconomic and development changes/issues
line 86 - was the questionnaire validated?
line 88 - not clear what method was used
line 92- what does "once qualified" mean?
line 96 - what does "double determination" mean? were all samples analysed twice?
line 114 - National Survey .... should this be capitalised?
line 122 - unified standard of data cleaning principle - can the authors provide a link to this for examination?
line 123 - English language - perhaps " any problematic record was rejected and returned to..."
line 126- reference or website for standard population?
For the statistical tests specifically how were "p's" determined, was there a correction for multiple t-tests, how did the authors check for co-linearity of independent variables?
line 141- what is "unqualified data"?
lines 144-145 - English language - please reword
p's in table one - to what do these refer and what statistical method was used tp calculate the p's of the methods discussed
under variable - last entry - " not answer" is grammatically incorrect
line 151 - how "weighted"
line 152- significant differences in "haemoglobin levels" ?
table 2 - as these are all derived from the same data set I believe they should have post-hoc analysis
and in table two - not always clear to hat the "p's" refer nor how calculated - perhaps the authors can describe how different "p's" calculated in a legend at the bottom of the table?
line 159 - how much lower?
line 160 - other BMI? can the authors be more specific please?
line 165 - not sure about the word "obviously" in a scientific manuscript - perhaps "as expected"?
line 171 appears to directly contradict line 156
in this results paragraph was do some results have "p's" while others do not?
line 176 - children does not need to be capitalised
table 4 - "p's" derivation not clear - please add legend at bottom of table to explain
line 204 - which year?
line 204 - perhaps consider "anemia was most strongly associated with female sex and age of..."
lines 217-218 - what are the relative variables?
line 222- perhaps change to "which was markedly improved COMPARED WITH THE 2002 CNNHS findings"
line 225-226 - what time frame?
line 232 - " THE Chinese..."
line 235 - which two studies?
line 247 - "grown" or improved?
lines 249-251 - please provide references/evidence to support these statements
line 253- is it an "epidemic" of anemia?
line267 - might hit sonly be true in these populations? in the USA obese individuals are often anemic
line 272 - reference incorrectly formatted
lines 284-286 - has BMI also increased during this time to support the hypothesis that nutrition in general has improved?
285 - so why are these policies not implemented here?
line 289 - please provide references to support these statements
lines 291-295 - I am not sure what point the authors are trying to make here?
Conclusion - see my introductory comments - this study presents associative evidence not causative - further evaluation with interventional studies is required to know what policies are actually improving anemia etc
Author Response
Response to Reviewer 2 Comments
Dear Reviewer:
Thank you very much for your comments on our paper submitted to IJERPH (Manuscript ID ijerph-436940).
We revised the manuscript and carefully proof-read the manuscript to minimize typographical, grammatical, and bibliographical errors. Here below is our description on revision according to your comments:
Point 1: This is an interesting study on the prevalence of anaemia and measurement of haemoglobin levels in two cohorts of Chinese children, and it is encouraging to see the improvement in haemoglobin levels and the fall in anaemia prevalence found in the study. The authors have done a good job analysing factors that are associated with anaemia and thus possible implications for future public health policies. Following are my general comments and then specific points line by line after that.
There are places where the English language used is somewhat confusing and I have highlighted some of those areas in the comments below with possible corrections for the authors to consider, but I would suggest that a review from someone with an excellent command of the English language might be helpful.
Response 1: We have asked for help to revise the English language, and the paper becomes more fluent and clearer.
Point 2: The abstract needs work - some language issues but also I find there are more results presented than the significance of the findings (ie discussion).
Response 2: The language issues and the content of abstract were revised.
Point 3: I find the data tables difficult to interpret especially the interpretation of the p's -sometimes it is difficult to know what the comparison being made is, and also as different statistical tests were used it is not clear from where the p is derived - ie what statistical test yielded that p?
Response 3: p values derived from different statistical tests in each table were explained in a legend at the bottom of the table.
Point 4: lastly, this is an observational study - thus although haemoglobin levels have improved and anaemia levels decreased causation has not been directly demonstrated - an intervention study is needed to prove that the public health policies rather than some other socio/cultural/economic factors had these effects - and furthermore which policies were most effective - this comment applies in particular to the authors' conclusion
Response 4: The conclusion was revised according to reviewer’s suggestion.
Point 5: line 15 - may I suggest "factors associated with anemia" - the statement as is sounds awkward
Response 5: The sentence has been changed to “factors associated with anemia.”
Point 6: line 15- the authors use American English throughout the manuscript (for example, anemia) other than "haemoglobin" - for consistency they should use "hemoglobin" - this error occurs through-out the manuscript
Response 6: All of the "haemoglobin" has been changed to "hemoglobin"
Point 7: line 16 - "between THE 2010, and after 48,610-children and adolescents
Response 7: The sentence has been changed to “from the 2010-2012 CNNHS, and after 48,610 children and adolescents aged 6-17 years in 2002 CNNHS were analyzed.”
Point 8: line 19- 135.2 is missing units of measurement, and "with THE prevalence"
Response 8: “THE” and the units of measurement of “g/L” have been added.
Point 9: line 20 - is there a "p" for this difference?
Response 9: A "p" for this difference has been added.
Point 10: line 32 - awkard language
Response 10: The sentence has been changed as “Anemia refers to a condition in which the number of red blood cells or their oxygen-carrying capacity is insufficient to meet physiological needs, and then adversely impact on health and social economic development [1, 2].”
Point 11: line 40 - "iron-deficiency"
Response 11: It has been changed to "iron-deficiency".
Point 12: line 41 "OTHER nutritional deficiencies"
Response 12: It has been changed to "other nutritional deficiencies ".
Point 13: line 44 - "there are lots" sounds like slang
Response 13: It has been changed to "There is much research".
Point 14: line 45 - no need for whilst at the start of the sentence
Response 14: “whilst” has been deleted.
Point 15: line 46 - please provide references to support these statements
Response 15: References have been provided to support these statements.
Point 16: line 49 - please provide references to support these statements
Response 16: References have been provided to support these statements.
Point 17: line 54 - please provide references to support these statements
Response 17: References have been provided to support these statements.
Point 18: line 55- please provide references to support these statements
Response 18: References have been provided to support these statements.
Point 19: line 57 - why did the anemia status remain unclear?
Response 19: The anemia status for Chinese residents except was clear, but children and adolescents aged 6-17y was not analyzed further.
Point 20: line 66 - please check grammar
Response 20: Grammar has been checked, and “were” was changed to “was”.
Point 21: line 73- what standards to define socioeconomic and development changes/issues.
Response 21: The standards that to divide the four strata of the survey was presented in the “Study Design and Participants”section.
Point 22: line 86 - was the questionnaire validated?
Response 22: The questionnaire was validated.
Point 23: line 88 - not clear what method was used
Response 23: The cyanmethaemoglobin method was used to measure hemoglobin levels, and it was presented clearly in the method after revision.
Point 24: line 92- what does "once qualified" mean?
Response 24: The sentence was changed to “After becoming qualified, they could take part in the hemoglobin measurement.”
Point 25: line 96 - what does "double determination" mean? were all samples analysed twice?
Response 25: The sentence was changed to “Each blood sample had two paralled determinations.”
Point 26: line 114 - National Survey.... should this becapitalised?
Response 26: “National Survey….” was be capitalized.
Point 27: line 122 - unified standard of data cleaning principle - can the authors provide a link to this for examination?
Response 27: The unified standard of data checking principle was added in the method section.
Point 28: line 123 - English language - perhaps " any problematic record was rejected and returned to..."
Response 28: The sentence was changed to “any problematic record was rejected and returned to”.
Point 29: line 126- reference or website for standard population?
Response 29: The reference for standard population was provided in the method section.
Point 30: For the statistical tests specifically how were "p's" determined, was there a correction for multiple t-tests, how did the authors check for co-linearity of independent variables?
Response 30: How were "p's" determined was explained in the method section and also in a legend under each table. Corrections were made for multiple tests between two groups. Correlation analysis and proc reg were used to check for co-linearity of independent variables.
Point 31: line 141- what is "unqualified data"?
Response 31: The "unqualified data" was judged by the data checking principle. To avoid misunderstanding, it was deleted in the sentence after revision.
Point 32: lines 144-145 - English language - please reword
Response 32: The sentence was reworded.
Point 33: p's in table one - to what do these refer and what statistical method was used tp calculate the p's of the methods discussed.
Response 33: p's in all tables was explained in a legend under the table after revision.
Point 34: under variable - last entry - "not answer" is grammatically incorrect
Response 34: "not answer" has been changed to “no response”.
Point 35: line 151 - how "weighted"
Response 35: Hemoglobin concentration and anemia rate analyses were adjusted for sample weights and the clustered survey design.
Point 36: line 152- significant differences in "haemoglobin levels" ?
Response 36: There was significant difference in "haemoglobin levels" between boys and girls.
Point 37: table 2 - as these are all derived from the same data set I believe they should have post-hoc analysis
Response 37: Post-hoc analysis was used to compare the two groups when conduct multiple tests.
Point 38: and in table two - not always clear to hat the "p's" refer nor how calculated - perhaps the authors can describe how different "p's" calculated in a legend at the bottom of the table?
Response 38: how different "p's" calculated and referred to in each table has been described in a legend at the bottom of the table.
Point 39: line 159 - how much lower?
Response 39: The level of hemoglobin In spring and summer was significantly lower than other seasons. the sentence was revised.
Point 40: line 160 - other BMI? can the authors be more specific please?
Response 40: other BMI has been revised more specifically.
Point 41: line 165 - not sure about the word "obviously" in a scientific manuscript - perhaps "as expected"?
Response 41: "obviously" was changed to “significantly”.
Point 42: line 171 appears to directly contradict line 156
Response 42: Although the participants aged 15–17 years had significantly higher hemoglobin level than other age groups, the prevalence of anemia was actually significantly higher than other age groups. Adolescents aged 15–17 years is also vulnerable to anemia.
Point 43: in this results paragraph was do some results have "p's" while others do not?
Response 43: We added the "p's" in sentence where is lack of.
Point 44: line 176 - children does not need to be capitalized
Response 44: Children wasn’t be capitalized after revision.
Point 45: table 4 - "p's" derivation not clear - please add legend at bottom of table to explain
Response 45: "p's" derivation was added at the bottom of table in a legend.
Point 46: line 204 - which year?
Response 46: The year of 2010-2012 was supplemented in the sentence.
Point 47: line 204 - perhaps consider "anemia was most strongly associated with female sex and age of..."
Response 47: The sentence was changed to” anemia was significantly associated with female”.
Point 48: lines 217-218 - what are the relative variables?
Response 48: The title was changed to” Determinants of anemia among Chinese children and adolescents in the CNNHS 2010–2012”.
Point 49: line 222- perhaps change to "which was markedly improved COMPARED WITH THE 2002 CNNHS findings"
Response 49: The sentence was revised according to reviewer’s suggestion.
Point 50: line 225-226 - what time frame?
Response 50: The time frame “during 2010-2012”was added into the sentence.
Point 51: line 232 - " THE Chinese..."
Response 51: The sentence was revised according to reviewer’s suggestion.
Point 52: line 235 - which two studies?
Response 52: In fact, they were two different surveys- the CNNHS 2010-2012 and the CNSSCH 2010. The sentence was revised.
Point 53: line 247 - "grown" or improved?
Response 53: The prevalence of anemia decreased significantly compared with the CNNHS 2002, children and adolescents living in poor rural areas showed the highest prevalence compared with cities and ordinary rural areas in the CNNHS 2010-2012.
Point 54: lines 249-251 - please provide references/evidence to support these statements
Response 54: References have been provided to support these statements.
Point 55: line 253- is it an "epidemic" of anemia?
Response 55: It was revised as “the prevalence of anemia”.
Point 56: line267 - might hit sonly be true in these populations? in the USA obese individuals are often anemic
Response 56: Some references were supplemented to discuss the relationship between BMI and anemia. Some studies found that the prevalence of anemia in overweight and obese was lower, while others found that obesity was significantly associated with iron deficiency.
Point 57: line 272 - reference incorrectly formatted
Response 57: The wrong format of reference was revised.
Point 58: lines 284-286 - has BMI also increased during this time to support the hypothesis that nutrition in general has improved?
Response 58: In this study, we didn’t analyze whether the BMI increased during this time. We deduced that the policy would improve the nutrition status of children and adolescents, and then increase the hemoglobin. But It could be interesting to research further.
Point 59: 285 - so why are these policies not implemented here?
Response 59: The policies were implemented in rural regions, but they maybe not insufficient to alleviate the prevalence of anemia for some children.
Point 60: line 289 - please provide references to support these statements
Response 60: References have been provided to support these statements.
Point 61: lines 291-295 - I am not sure what point the authors are trying to make here?
Response 61: This section was to present the limitations of this survey. It was reworded to be clearer than before.
Point 62: Conclusion - see my introductory comments - this study presents associative evidence not causative - further evaluation with interventional studies is required to know what policies are actually improving anemia etc
Response 62: The conclusion was revised according to reviewer’s suggestion.

Round 2
Reviewer 1 Report
The majority of the suggestions have been addressed.
However the issue of those with high Hb level is still confusing. The authors state 30836 participant had Hb levels >recommendations (line 199). However, I was not asking the number with adequate Hb, but those with Hb levels above the optimal range, i.e. those with too high Hb levels. Please could you add this information to your results and discussion.
Author Response
Dear Reviewer:
Thank you very much for your comments on our paper submitted to IJERPH (Manuscript ID ijerph-436940).
We revised the manuscript and carefully proof-read the manuscript to minimize typographical, grammatical, and bibliographical errors. Here below is our description on revision according to your comments:
Point 1: Comments and Suggestions for Authors: The majority of the suggestions have been addressed.
However the issue of those with high Hb level is still confusing. The authors state 30836 participant had Hb levels >recommendations (line 199). However, I was not asking the number with adequate Hb, but those with Hb levels above the optimal range, i.e. those with too high Hb levels. Please could you add this information to your results and discussion.
Response 1:Thanks very much for your comprehensive and profound suggestions, which reminding us to concern those too high Hb values. These values were deserved for considering. But after carefully thinking, we are not going to report this result or discuss them in this manuscript. Therefore, we beg for your understanting and the reasons are described as follow:
(1) There is a uniform standard cut-off points for anemia from WHO, but there is no specific cut-off points for high Hb levels, especially for children and adolescents.
(2) The geography of china is complicated, and the Hb level varies a lot as different altitude. According to some chinese references, the mean Hb level of population from plateau regions could be 180-200 g/L, which still belongs to a physiologically high status. Thus it’s difficult for us to judge the high Hb level or too high Hb level of Chinese children and adolescents.
(3) Combined with the geography and population, the survey panel of CNNHS
suggested that Hb values higher than 230 g/L would be returned to the monitoring site
for re-inspection. In the present study, there were 36 children whose Hb level were
higher than 200 g/L, and only 6 of them (0.02%) were higher than 230 g/L. This was a
extremely small ratio. Afer re-inspection, these values were examined correctly.
These children and adolescents were students from regions with high altitude, and
they were basically healthy. Therefore, the high Hb values were judged to
physiological high ones.
(4) One of the focuses of this investigation was to assess anemia and analyze the factors associated with anemia, and then to carry out intervention studies to alleviate anemia. Althogh the high Hb values needed to pay attention, we plan to analyze and report them further in other studies.
In conclusion, we are not going to report or discuss these high Hb levels in this manuscript. We beg for reviewer’s forgiveness. For all this, we are so appreciated for these valuable suggestions on this paper.

Reviewer 2 Report
Overall the manuscript is improved. The following overall points remain which I feel must be addressed - specific points follow afterwards.
1) The English remains awkward in quite a few places - I have suggested corrections in some places, but the whole manuscript needs to be reviewed
2) the statistics still need work - it is still not always clear what method was used to calculate the p's in certain places (not in all places), and if there were a priori hypotheses - or if not what type of post hoc analysis was conducted - for example, was it Bonferroni? etc
Specific points:
Line 13-14 - English - may I suggest -This study assessed hemoglobin levels...and analysed.
lines 15-17 - still needs work on the English - not clear
line 25 - what exactly is "mild" - is it really mild? why not just say public health problem?
line 33- may I suggest " physiological need. Anemia adversely impacts health and social economic development; children and women are particularly vulnerable.
line 35 - not sure "deter" is the correct word - consider "detrimentally affect"
line 38- remove "the" before developed and add countries after developed
line 45- replace "most of them" (grammatically incorrect" with "much of it"
line 47 - is data limited in all parts of the world or just in China?
line 52 - comma needed after "Chinese citizens"
line 54- remove "the" before Chinese
line 56 - economy HAS
line 59 - why unclear? because it was not measured?
line 63- unclear- may I suggest - "potential factors associated with anemia."
line 66 - unclear - may I suggest - "Data of children....CNN2010-2012 survey
line 121- 5-11 years old
line 134 - which ARE
line 136 - Hemoglobin level data of each monitoring site.....
lines 138-140 - hard to follow - please reword
line 146- English- Hemoglobin concentration data
line 152 - what kind of post hoc analysis?
line 166 - "that had their hemoglobin measured"
Table 1 - Total N of Han population and other population does not add up to 33,015
Table 1 - why is there a line between South and Season?
Table 1 - needs post hoc - and specific correction used
line 178 - hypothesis?
lines 179-182 - please be specific - how much lower and significant? how much higher and significant?
Table 2 - KW test is non-parametric ANOVA - not clear to what the p values refer? as there are only two categories so why not use Mann-Whitney? what specific post-hoc analysis was used?
line 199- there were 30836 participants WHO had hemoglobin levels above recommendations - what does this mean? were their hemoglobins pathologically elevated to suggest polycythemia?
line 204 - how much higher - please be specific
Table 3 - what type of post hoc analysis?
line 219 - please remove "obviously"
line 242- female GENDER (or sex)
line 275 - how much higher?
line 274 - remove "of" before hemoglobin
line 281 - remove "of" before hemoglobin
line 286-287 - English needs work
line 286 - socioeconomic of what? please specify
line 343 - Intervention studies - studies of intervention incorrect
Author Response
Dear Reviewer:
Thank you very much for your comments on our paper submitted to IJERPH (Manuscript ID ijerph-436940).
We revised the manuscript and carefully proof-read the manuscript to minimize typographical, grammatical, and bibliographical errors. Here below is our description on revision according to your comments:
Point 1: Overall the manuscript is improved. The following overall points remain which I feel must be addressed - specific points follow afterwards.
1) The English remains awkward in quite a few places - I have suggested corrections in some places, but the whole manuscript needs to be reviewed
Response 1: English has been reviewed and corrected carefully this time, including places where reviewer pointed out and the whole manuscript.
Point 2: 2) the statistics still need work - it is still not always clear what method was used to calculate the p's in certain places (not in all places), and if there were a priori hypotheses - or if not what type of post hoc analysis was conducted - for example, was it Bonferroni? Etc
Response 2: The statistical analysis was described specifically after revised. The methods for post hoc analysis were also supplemented into the part of “Statistical Analysis”. p's was clearly reported in results. Results for multiple comparisons were added in the table.
Point 3: Specific points: Line 13-14 - English - may I suggest -This study assessed hemoglobin levels...and analysed.
Response 3: The sentence has been changed to “This study assessed hemoglobin levels and anemia status of Chinese children and adolescents from Chinese National Nutrition and Health Survey (CNNHS) in 2010–2012 and analyzed the factors associated with anemia.”, according to reviewer’s suggestion.
Point 4: lines 15-17 - still needs work on the English - not clear
Response 4: The sentence has been changed to “The hemoglobin concentration and prevalence of anemia for children and adolescents aged 6-17 years from both
CNNHS 2010-2012 and CNNHS 2002 were analyzed”.
Point 5: line 25 - what exactly is "mild" - is it really mild? why not just say public health problem?
Response 5: "mild" was deleted and the sentence has been changed to “it remains a public health problem in this population”, according to reviewer’s suggestion.
Point 6: line 33- may I suggest "physiological need. Anemia adversely impacts health and social economic development; children and women are particularly vulnerable.
Response 6: The sentence has been changed to “physiological need. Anemia adversely impacts health and social economic development; children and women are particularly vulnerable”, according to reviewer’s suggestion.
Point 7: line 35 - not sure "deter" is the correct word - consider "detrimentally affect"
Response 7: "deter" was deleted and the word has been changed to “detrimentally affect”, according to reviewer’s suggestion.
Point 8: line 38- remove "the" before developed and add countries after developed
Response 8: "the" before developed was removed and "countries" was added after developed, according to reviewer’s suggestion.
Point 9: line 45- replace "most of them" (grammatically incorrect" with "much of it"
Response 9: "most of them" were replaced with “much of it”, according to reviewer’s suggestion.
Point 10: line 47 - is data limited in all parts of the world or just in China?
Response 10: The data was limited worldwide, including China.
Point 11: line 52 - comma needed after "Chinese citizens"
Response 11: comma was added after "Chinese citizens", according to reviewer’s suggestion.
Point 12: line 54- remove "the" before Chinese
Response 12: "the" before Chinese was removed, according to reviewer’s suggestion.
Point 13: line 56 - economy HAS
Response 13: "HAS" was added after economy, according to reviewer’s suggestion.
Point 14: line 59 - why unclear? because it was not measured?
Response 14: The sentence has been changed to “However, the anemia status of school-aged children in the 2010-2012 CNNHS has not been assessed”.
Point 15: line 63- unclear- may I suggest - "potential factors associated with anemia."
Response 15: The sentence has been changed to “potential factors associated with anemia.”.
Point 16: line 66 - unclear - may I suggest - "Data of children....CNN2010-2012 survey
Response 16: The sentence has been changed to "Data of children and adolescents aged 6–17 years were drawn from CNNHS 2010–2012"
Point 17: line 121- 5-11 years old
Response 17: The sentence has been changed to " for children aged5-11 years:"
Point 18: line 134 - which ARE
Response 18: The sentence has been changed to "which ARE".
Point 19: line 136 - Hemoglobin level data of each monitoring site.....
Response 19: The sentence has been changed to "Hemoglobin level data of each monitoring site.....".
Point 20: lines 138-140 - hard to follow - please reword
Response 20: The part of "Data Check" has been reworded clearly this time: The unified standard of data check principle was mainly formulated on the following aspects: (1) connecting the hemoglobin database to the basic information base to check if there was duplicate information; (2) checking whether the difference from the hemoglobin values of parallel samples was less than 20%; (3) according to the absorbance value of spectrophotometer, calculating the hemoglobin level and then comparing with the reported values.
Point 21: line 146- English- Hemoglobin concentration data
Response 21: The sentence has been changed to "Hemoglobin concentration data".
Point 22: line 152 - what kind of post hoc analysis?
Response 22: The statistical analysis was described specifically after revised. The methods for post hoc analysis were also supplemented into the part of “Statistical Analysis”. As for Hemoglobin levels, the Dwass-Steel-Critchlow-Fligner (DSCF) method was used to conduct post hoc analysis. As for the rates of anemia, Rao-scott test was also used for multiple comparisons. Bonferroni correction was applied to obtain the adjusted p values for multiple tests.
Point 23: line 166 - "that had their hemoglobin measured"
Response 23: The sentence has been changed to "that had their hemoglobin measured ".
Point 24: Table 1 - Total N of Han population and other population does not add up to 33,015
Response 24: 33015 subjects had their hemoglobin measured. When analyzed for sub-groups, 262 subjects were missing for their ethnicity information, so the total N of Han population and other population was 32753. In order to avoid misunderstanding, slight changes were made and “Total” in the second line was removed.
Point 25: Table 1 - why is there a line between South and Season?
Response 25: The line between South and Season was deleted.
Point 26: Table 1 - needs post hoc - and specific correction used
Response 26: Post hoc was conducted and Bonferroni correction was applied to obtain the adjusted p values for multiple tests.
Point 27: line 178 - hypothesis?
Response 27: Post hoc was conducted and the specific significance levels were supplemented in this part.
Point 28: lines 179-182 - please be specific - how much lower and significant? how much higher and significant?
Response 28: The results were described specifically after revised: Post-hoc analysis showed that participants aged 15–17 years had significantly higher hemoglobin level than those aged 12-14 years (p<0.0001), 9-11 years (p<0.0001) and 6-8 years (p<0.0001). The hemoglobin level in poor rural areas was significantly lower than that in large cities (p<0.0001), small and medium-sized cities (p<0.0001) and ordinary rural area (p<0.0001). Children living in northern areas had significantly higher hemoglobin level than those living in southern areas (p<0.0001). The hemoglobin level in winter was significantly higher than that in autumn (p<0.0001), spring (p<0.0001) and summer (p<0.0001).
Point 29: Table 2 - KW test is non-parametric ANOVA - not clear to what the p values refer? as there are only two categories so why not use Mann-Whitney? what specific post-hoc analysis was used?
Response 29: Wilcoxon (Mann-Whitney) test was used for comparing difference between two groups. The Kruskal-wallis test was used for comparing difference among three or more groups, and then followed by the Dwass-Steel-Critchlow-Fligner (DSCF) method to conduct post hoc analysis.
Point 30: line 199- there were 30836 participants WHO had hemoglobin levels above recommendations - what does this mean? were their hemoglobins pathologically elevated to suggest polycythemia?
Response 30: In order to avoid misunderstanding, this sentence was removed.
Point 31: line 204 - how much higher - please be specific
Response 31: The sentence was revised more specifically: In spring and summer, the prevalence of anemia was above 10.0%, which was significantly higher than that in autumn (all p values<0.0001) and winter (all p values<0.0001).
Point 32: Table 3 - what type of post hoc analysis?
Response 32: Rao-scott test was used for post hoc analysis and Bonferroni correction was applied to obtain the adjusted p values.
Point 33: line 219 - please remove "obviously"
Response 33: "obviously" was removed according to reviewer’s suggestion.
Point 34: line 242- female GENDER (or sex)
Response 34: The sentence has been changed to “female gender".
Point 35: line 275 - how much higher?
Response 35: The sentence has been changed to “The mean hemoglobin concentration of boys was significantly higher than that of girls in total and in all the sub-groups in this survey except for children aged 6-11 years".
Point 36: line 274 - remove "of" before hemoglobin
Response 36: "of" before hemoglobin was removed.
Point 37: line 281 - remove "of" before hemoglobin
Response 37: "of" before hemoglobin was removed.
Point 38: line 286-287 - English needs work
Response 38: This part was changed to "Although Chinese economy has grown rapidly in both urban and rural areas in recent years [14], living conditions in urban remain better than rural areas [27, 28]."
Point 39: line 286 - socioeconomic of what? please specify
Response 39: The sentence has been changed to "Chinese economy…..".
Point 40: line 343 - Intervention studies - studies of intervention incorrect
Response 40: The sentence has been changed to “Intervention studies are needed to prove…. ".
